# Spread of endemic SARS-CoV-2 lineages in Russia before April 2021

**Galya V. Klink**[1], **Ksenia R. Safina**[2], **Sofya K. Garushyants**[1], **Mikhail Moldovan**[2], **Elena Nabieva**[2], **Andrey B. Komissarov**[3], **Dmitry Lioznov**[3,4], **Georgii A. Bazykin**[1,2]*, **The CoRGI (Coronavirus Russian Genetic Initiative) Consortium**[¶]

**1** A.A. Kharkevich Institute for Information Transmission Problems of the Russian Academy of Sciences, Moscow, Russia, **2** Skolkovo Institute of Science and Technology (Skoltech), Moscow, Russia, **3** Smorodintsev Research Institute of Influenza, Saint Petersburg, Russia, **4** First Pavlov State Medical University, Saint Petersburg, Russia

¶ https://corgi.center/en/ (see the list of consortium members in S1 File)
* g.bazykin@skoltech.ru

**Data Availability Statement:** All relevant data are within the manuscript and its Supporting Information files.

## Abstract

In 2021, the COVID-19 pandemic was characterized by global spread of several lineages with evidence for increased transmissibility. Throughout the pandemic, Russia has remained among the countries with the highest number of confirmed COVID-19 cases, making it a potential hotspot for emergence of novel variants. Here, we show that among the globally significant variants of concern that have spread globally by late 2020, alpha (B.1.1.7), beta (B.1.351) or gamma (P.1), none have been sampled in Russia before the end of 2020. Instead, between summer 2020 and spring 2021, the epidemic in Russia has been characterized by the spread of two lineages that were rare in most other countries: B.1.1.317 and a sublineage of B.1.1 including B.1.1.397 (hereafter, B.1.1.397+). Their frequency has increased concordantly in different parts of Russia. On top of these lineages, in late December 2020, alpha (B.1.1.7) emerged in Russia, reaching a frequency of 17.4% (95% C.I.: 12.0%-24.4%) in March 2021. Additionally, we identify three novel distinct lineages, AT.1, B.1.1.524 and B.1.1.525, that have started to spread, together reaching the frequency of 11.8% (95% C.I.: 7.5%-18.1%) in March 2021. These lineages carry combinations of several notable mutations, including the S:E484K mutation of concern, deletions at a recurrent deletion region of the spike glycoprotein (S:Δ140–142, S:Δ144 or S:Δ136–144), and nsp6:Δ106–108 (also known as ORF1a:Δ3675–3677). Community-based PCR testing indicates that these variants have continued to spread in April 2021, with the frequency of B.1.1.7 reaching 21.7% (95% C.I.: 12.3%-35.6%), and the joint frequency of B.1.1.524 and B.1.1.525, 15.2% (95% C.I.: 7.6%-28.2%). Although these variants have been displaced by the onset of delta variant in May-June 2021, lineages B.1.1.317, B.1.1.397+, AT.1, B.1.1.524 and B.1.1.525 and the combinations of mutations comprising them that are found in other lineages merit monitoring.

**Funding:** G.A.B.; grant 20-54-80014; Russian Foundation for Basic Research; https://www.rfbr.ru/rffi/eng. The funders had no role in study design, data collection and analysis, decision to publish, or preparation of the manuscript.

## Introduction

Continuing evolution of SARS-CoV-2 in humans leads to emergence of new variants with novel epidemiological and/or antigenic properties. In spring 2020, the S:D614G change has spread globally due to its fitness advantage [1,2]. Subsequently, a number of lineages that were designated as variants of concern (VOCs) by the World Health Organisation [3], including alpha (B.1.1.7) first sampled in Great Britain in September, beta (B.1.351) first sampled in South Africa in October, gamma (P.1) first sampled in Brazil in December and delta (B.1.617.2) fist sampled in India in October, were shown to be associated with increased transmissibility [4–6]. These variants are characterized by overlapping sets of changes in spike receptor-binding domain which affect ACE2 binding and antibody recognition, as well as other changes with demonstrated functional and antigenic effects. Emergence of SARS-CoV-2 variants with evidence for change in transmissibility, and possibly other properties, highlights the importance of continued surveillance of novel variants. In particular, locally arising variants that grow in frequency over time may suggest a transmission advantage, although such an increase may also occur by chance [7].

Here, we show that before spring 2021, the outbreak in Russia was characterized by a spread of two lineages, B.1.1.317 and B.1.1.397+, which were highly prevalent in Russia but rarely appeared in non-Russian samples. We trace the accumulation of sequential mutations in the evolution of these lineages, and single out the spike mutations that were followed by bursts in frequency. If the frequency increase of B.1.1.317 and B.1.1.397+ has been indeed driven by changes in the intrinsic properties of the virus rather than by epidemiology, it is these mutations in spike that most likely have led to this increase, although importantly we lack direct transmission data to verify causality. We also describe three variants that were characterized by a rapid increase in frequency and combinations of important spike mutations, including the E484K mutation of concern.

## Materials and methods

### Dataset preparation

1,060,545 sequences of SARS-CoV-2 were downloaded from GISAID on April 15, 2021 (Supplementary File 2) and aligned with MAFFT [8] v7.45324 against the reference genome Wuhan-Hu-1/2019 (NCBI ID: MN908947.3) with--addfragments--keeplength options using default parameters of MAFFT for handling nonspecific matches. 100 nucleotides from the beginning and from the end of the alignment were trimmed. After that, we excluded sequences (1) shorter than 29,000 bp, (2) with more than 3,000 (for Russian sequences) or 300 (for all other countries) positions of missing data (Ns), (3) excluded by Nextstrain [9], (4) in non-human animals, (5) with a genetic distance to the reference genome more than four standard deviations from the epi-week mean genetic distance to the reference, or (6) with incomplete collection dates. As our focus was on the spread of lineages in Russia, and since Russia is relatively poorly sampled, we chose a less stringent threshold at step (2) for Russian compared to non-Russian sequences in order to keep more Russian data in the dataset. The average length of sequences in the alignment was 29687.29 bp, and the average Hamming distance from the reference was 0.000799. To this dataset, we added the 1,645 Russian sequences described in our earlier study [10] and 344 samples produced by the CoRGI consortium which had not been yet available in GISAID on April 15 (all these samples have been deposited to GISAID since then). The final dataset consisting of 830,249 sequences, including 4,487 Russian samples, was then annotated by the PANGOLIN package (v2.3.8). For the categories in Fig 1, we selected Pango lineages represented by at least 100 Russian samples (there were six such

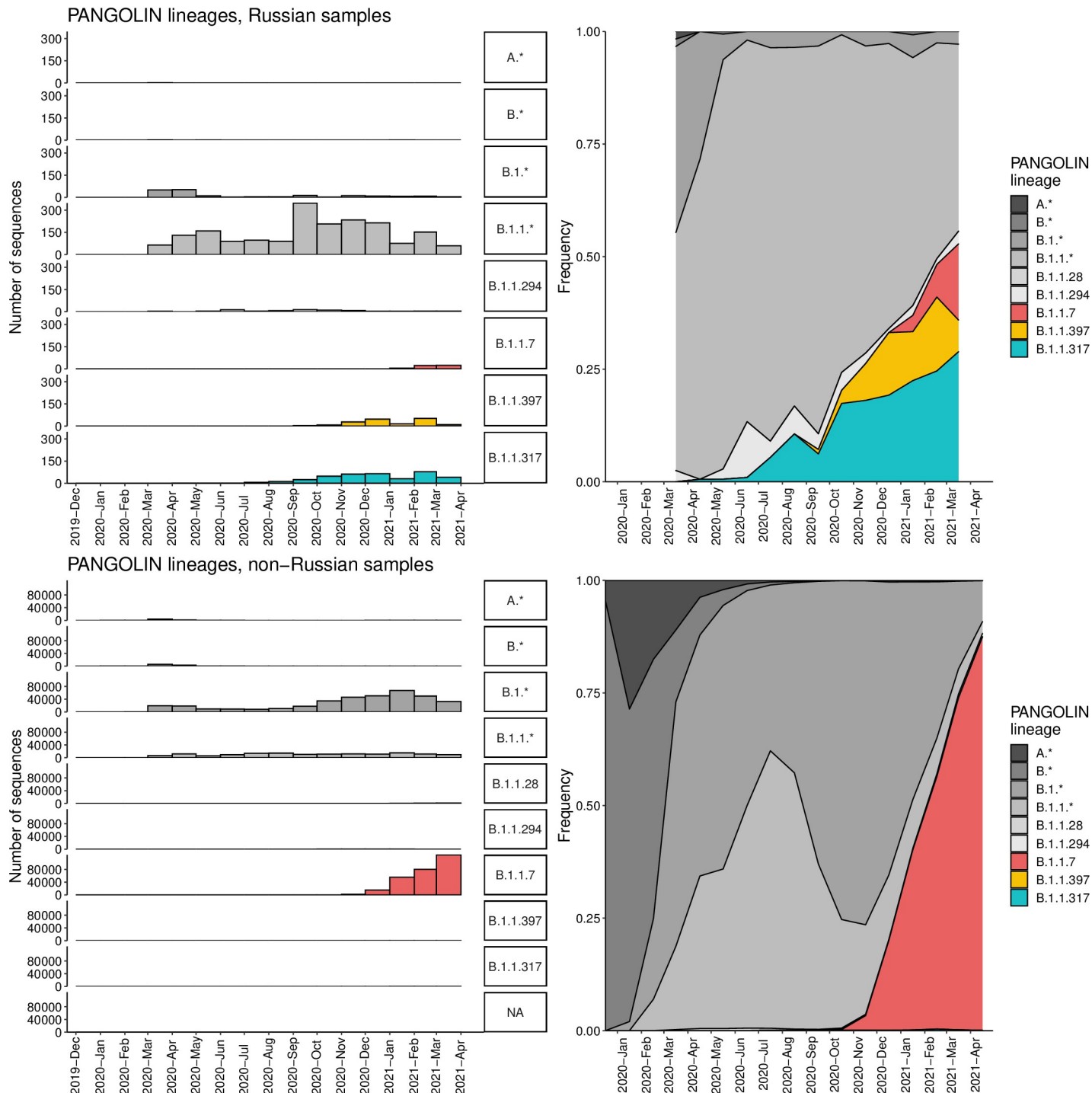

**Fig 1. Dynamics of Pango lineage frequencies in Russia (top row) and among the non-Russian samples in GISAID (bottom row).** Asterisks in Pango lineage designations correspond to pooled sets of lineages of that hierarchy level, except those listed in other categories; e.g., B.1.1.* includes B.1.1 and B.1.1.6 but not B.1.1.7 or B.1.1.317. Samples are split into one month bins.

lineages: B.1, B.1.1, B.1.1.397, B.1.1.28, B.1.1.317 and B.1.1.294), added the two earliest lineages A and B and the VOC lineage B.1.1.7, and aggregated A, B, B.1 and B.1.1 with lineages nested in them except those included in other categories into A.*, B.*, B.1.* and B.1.1.*, respectively.

## Data analysis

To trace the frequency dynamics of mutations, for each non-reference nucleotide at each position in each month, we calculated the fraction of samples with this nucleotide among all samples where this position contained an ambiguous nucleotide. We did this separately for the Russian and non-Russian samples, and selected changes with frequency above 5% in S protein or above 10% in other proteins in February-March 2021 (a total of 461 sequences) in Russian samples for consideration. Calculations were performed with custom Perl and R scripts. Wilson score intervals were estimated using Hmisc R package [11], and results were visualized with the ggplot2 package [12] of R language [13]. Upset plots were built with the UpSetR package [14] for R. Logistic growth models were fit to frequencies of each variant averaged across 14-days sliding windows with nls() function of R language [13]. Only windows with a total number of samples > 20 were taken into account. Confidence intervals for estimated parameters were obtained with confint2 function from nlstools R package [15], similar to [16]. Results were visualized with R packages ggplot2 [12], tidyverse [17] and gridExtra [18].

Variants of concern (VOC) or variants of interest (VOI) were determined according to the World Health Organization [3], and mutations of concern or mutations of interest were determined according to outbreak.info [19].

To estimate positive selection, we employed MEME and FEL models implemented in the HyPhy package [20,21]. For this analysis, we added Russian sequences with incomplete collection dates to the main dataset, which resulted in 5006 Russian sequences. The tree for the selection analysis was built upon the whole-genome alignment of Russian sequences with the RAxML package v.8.0.26 (model GTRCAT) [22].

## Mapping residues onto the S-protein structure

To visualize spike mutations, we utilized two different spike protein structures, corresponding to the S-protein in a complex bound with 4A8 (PDB ID: 7c2l) or with P5A-1B9 (PDB ID: 7czx) antibody. The NTD antibody binding epitope is defined as in [23]. The two structures were structurally aligned and visualized with Open-Source PyMOL [24].

## PCR data

Community-based PCR tests aimed at detection of S:Δ69–70 and nsp6:Δ106–108 deletions were performed for 739 samples. We further analyzed only those samples for which both tests were performed and produced unambiguous results. There were 269 such samples from 22 regions (including 170 from Saint Petersburg, 43 from Sverdlovsk Oblast, and 12 from Leningrad Oblast) obtained between February-April, 2021. For S:Δ69–70 detection, we used the Yale69/70del RT-PCR assay described elsewhere [25]. For nsp6:Δ106–108 detection, we used a newly designed RT-PCR assay. 133 of these 269 samples were also sequenced (sequence data made available through GISAID); for 126 of them (94.7%), the results on the presence of S:Δ69–70 and nsp6:Δ106–108 were consistent between the NGS and PCR data, indicating that our PCR tests are highly specific.

## Results

### High-frequency variants in Russia

We analyzed 4,487 SARS-CoV-2 sequences with known sampling dates obtained in Russia between February 25, 2020—March 28, 2021 and deposited to GISAID [26]. The vast majority of samples over this period came from several genomic surveillance programs which were not

targeted towards particular variants, although representation of Russia's regions varied with time.

Before the spread of Delta in spring 2021, the SARS-CoV-2 diversity in Russia has been predominated by the B.1.1 Pango lineage which was frequent in Europe, as well as lineages descendant from it [27] (Fig 1). Three B.1.1-derived lineages with the highest prevalence in Russia in the beginning of 2021 were B.1.1.7 first sampled in Russia on December 25, 2020, as well as two other lineages, B.1.1.317 and B.1.1.397, that appeared in Russia in April and July 2020 respectively. B.1.1.317 was first sampled in Vietnam on March 27, 2020 [28]; within Russia, it was first sampled on April 5, 2020 in Moscow, spreading across the country throughout 2020 (Fig 2). B.1.1.397 was first sampled in the Krasnoyarsk Region of Russia on July 22, 2020. By summer 2020, both B.1.1.317 and B.1.1.397 were frequent throughout Russia (Fig 2).

To find the non-reference amino acid variants that gained in frequency in Russia, we selected the positions at which the mean frequency of the non-reference variant in Russian samples exceeded 5% (for the spike) or 10% (for other proteins) in February-March 2021. We found 21 such positions in spike and 21 such positions in other proteins. Among these changes, two (RdRp:P323L and S:D614G) were fixed early in the global evolution of SARS--CoV-2; other two (N:R203K and N:G204R) are the lineage-defining mutations of B.1.1.

The frequency dynamics of the derived variants at the remaining 38 positions is shown in Fig 3. These include the mutations characterizing the B.1.1.7 variant which has been increasing in frequency in Russia since January 2021 (Fig 1), as well as some of the other globally spreading mutations of concern or interest, including the E484K mutation in spike. However, at many of these sites, the non-reference variants were rare outside Russia (Fig 3). Most of these variants showed similar temporal dynamics in Moscow and St. Petersburg regions, as well as in the European and Asian parts of Russia (S1 Fig), indicating that their increase in frequency is not a result of sampling bias.

We aimed to identify the high-frequency variants carrying these mutations. Many of these sites were highly homoplasic, and overall we found the resulting phylogenies not to be robust. Instead, we defined the most frequent variants composed of these mutations, independent of the alleles at other sites (Fig 4).

We considered the allele combinations that were most frequent in Russia in February-March 2021, and noticed that they belonged to several nested sets. The most frequent combination (99 out of the 461 samples) carried the N:A211V mutation which is characteristic of the B.1.1.317 Pango lineage; the second most frequent combination carried the S:D138Y and ORF8:V62L combination of mutations which are characteristic of the B.1.1.397 lineage; the third combination carried the set of characteristic mutations of B.1.1.7.

Still, there was no one-to-one correspondence between the frequent combinations of mutations and Pango lineages. For example, while the most frequent variant carrying S:M153T also included N:M234I, S:D138Y and ORF8:V62L (column 2 in Fig 4) and was classified as Pango lineage B.1.1.397, the variants carrying S:M153T alone, the S:M153T+N:M234I combination and the S:M153T+N:M234I+S:N679K combination were also frequent (columns 4, 9 and 5 in Fig 4 respectively) but were classified by PANGOLIN as other lineages (B.1.1, B.1.1.141, B.1.1.28 and others). Similarly, while B.1.1.317 is defined by the N:A211V mutation, the frequency of the variant carrying this mutation alone was relatively low (column 12 in Fig 4), while most samples carrying it also carried 8 additional high-frequency mutations, including four potentially important changes in spike (Q675R+D138Y+S477N+A845S; column 1 in Fig 4). The frequencies of such "non-canonical" combinations of mutations increased throughout 2020–2021 (Fig 5).

Finally, we observe three high-frequency combinations of mutations, including the S: E484K mutation of concern as well as other mutations of interest according to outbreak.info

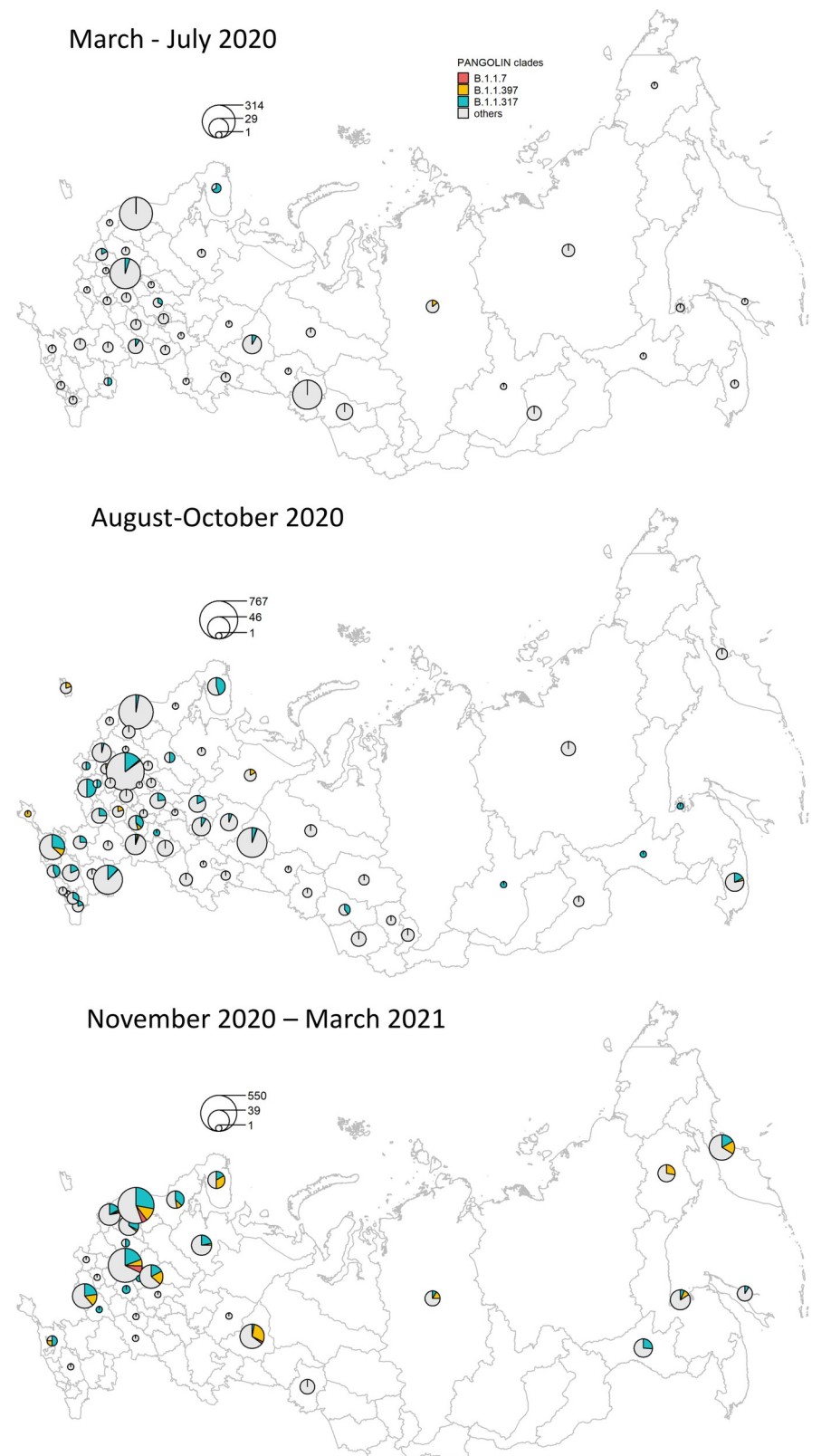

**Fig 2. The spatio-temporal distribution of lineages that were frequent in Russia before April 2021.**

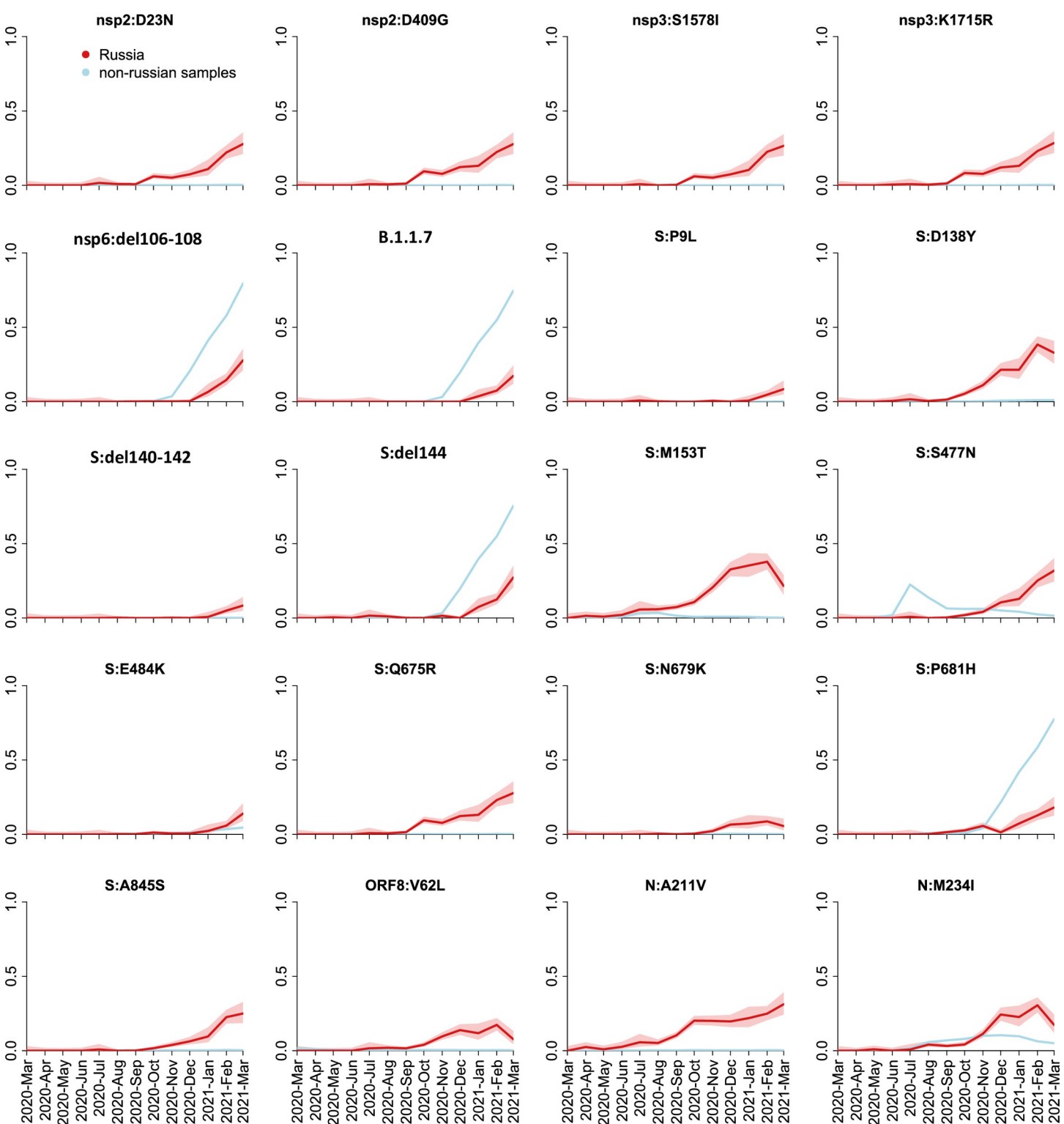

**Fig 3. Frequency dynamics of SARS-CoV-2 amino acid changes.** Plots represent changes in frequency over time for the non-reference amino acid mutations that reached frequencies above 10% (5% for the S protein) among the 461 Russian samples obtained in February-March 2021. The frequency of B.1.1.7 is represented by the mutation nsp3:I1412T; deletions nsp6:Δ106–108 and S:Δ144 and substitution S:P681H which are a part of B.1.1.7 as well as other lineages are shown separately; the remaining 14 mutations such that >70% of samples carrying them belonged to B.1.1.7 are not shown. Changes in frequency in Russian (red) and non-Russian (blue) samples are shown in one-month time intervals. Shaded areas show 95% confidence intervals (Wilson score intervals).

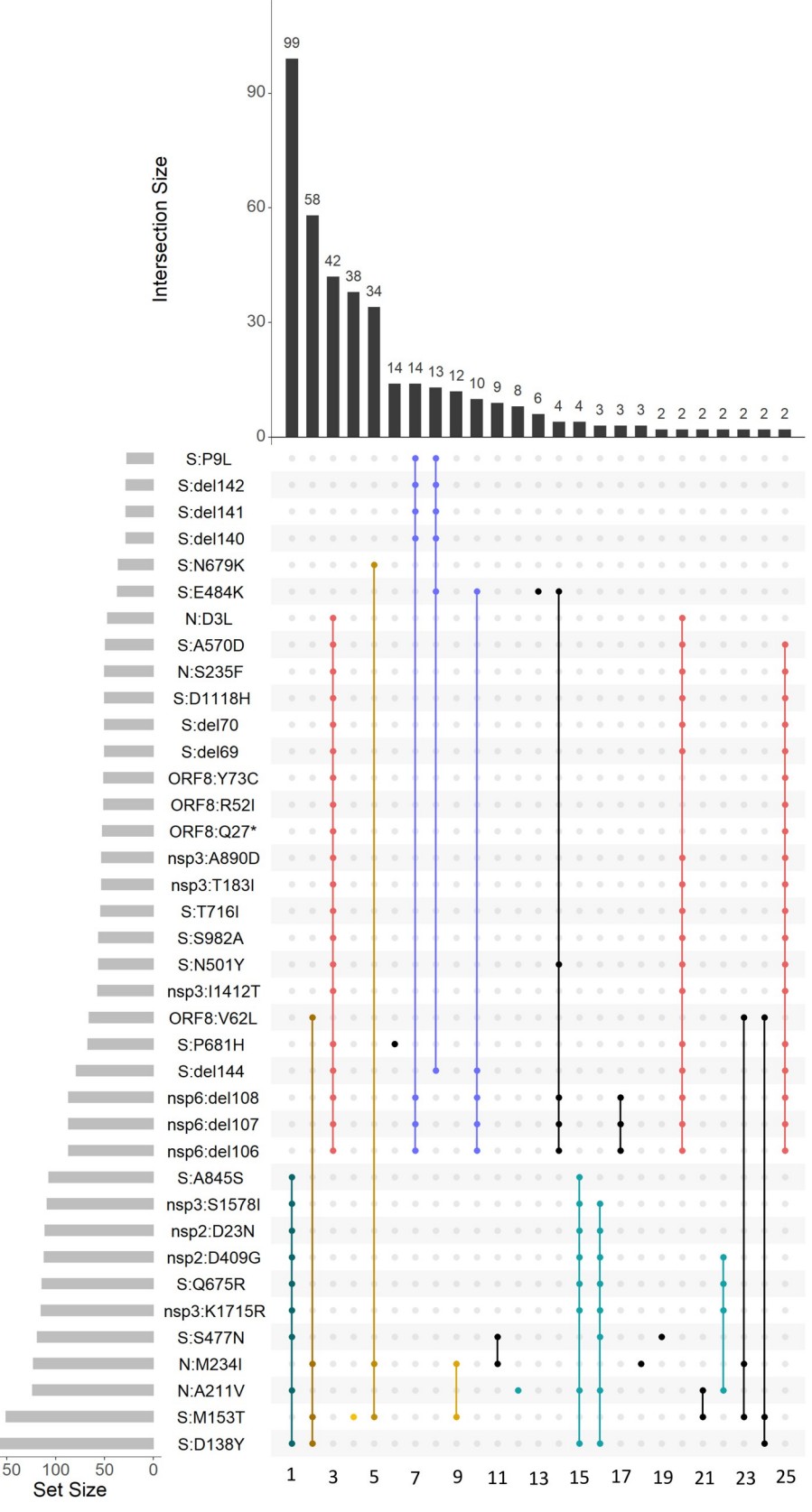

**Fig 4. Variants with high frequencies in Russia in February-March.** Horizontal rows represent all positions with non-reference alleles at frequency above 5% (for spike protein) or 10% (for other proteins) in Russia in February-March. Columns represent all observed combinations of these variants that included 2 or more samples, with black or colored dots indicating the presence of the non-reference variant. Colors dots represent the variants that are discussed in the text, with the same color coding as in Figs 1, 2 and 5; blue color corresponds to variants B.1.1.524 (column 7), AT.1 (column 8) and B.1.1.525 (column 10).

[19] (notably S:Δ140–142, S:Δ136–144 and nsp6:Δ106–108, also referred to as ORF1a:Δ3675–3677; columns 7, 8 and 10 in Fig 4). These combinations have later been designated as Pango lineages B.1.1.524, AT.1 and B.1.1.525 (for columns 7, 8 and 10 in Fig 4, respectively).

## Frequency dynamics of the variants prevalent in Russia

To describe how the frequency of the variants has changed over time, and to single out the possible candidate individual mutations with potential for effect on viral fitness, we fit the logistic growth model for the 10 most-frequent combinations of mutations and for N:A211V (which was the 12th most-frequent combination), and compared the dynamics of nested combinations with each other (Figs 6–8).

The variant carrying just the N:A211V change (largely coincident with the B.1.1.317 Pango lineage) has increased in frequency since the start of the epidemic in Russia. However, since fall 2020, it was being displaced by the variant with 8 additional mutations, including four in spike: Q675R+D138Y+S477N+A845S. When the logistic growth model was fit to the N:A211V variant alone, it demonstrated a modest growth (Fig 6A); however, its combination with S:Q675R+D138Y+S477N+A845S demonstrated a much more rapid frequency increase, with the estimated daily growth rate of 1.93% (95% CI: 1.8%–2.06%; Fig 6B). While this led to a frequency increase of the N:A211V mutation independent of the background (Fig 6C), this suggests that the frequency increase was characteristic of the S:Q675R+D138Y+S477N+A845S combination rather than the N:A211V change defining the B.1.1.317 Pango lineage.

By contrast, the frequency of the S:M153T mutation grew independently of the presence of other mutations from our list (Fig 7). Indeed, the estimated growth rates were comparable when the S:M153T mutation occurred alone or in combination with N:M234I, N:M234I+S:N679K, or N:M234I+S:D138Y, and all these combinations were still frequent many months after they had originated (Figs 5 and 7).

Finally, the five remaining variants which also reached high frequency in February-March carried unnested, although partially overlapping sets of mutations. These included B.1.1.7 (Fig 8A); a variant carrying the S:P681H mutation of interest in the absence of other high-frequency mutations (Fig 8B); as well as three novel variants carrying the following combinations of mutations: (i) nsp6:Δ106–108+S:P9L+S:Δ140–142 (which recently got the B.1.1.524 Pango designation; Fig 8C); (ii) S:P9L+S:Δ136–144+S:E484K (which recently got the AT.1 Pango designation; Fig 8D); and (iii) nsp6:Δ106–108+S:Δ144+S:E484K (which recently got the B.1.1.525 Pango designation; Fig 8E). Variants B.1.1.524, AT.1 and B.1.1.525 were only observed in 13–14 samples each, but are of interest because this constitutes an appreciable fraction of samples obtained in February-March (3.0%, 2.8% and 2.6% respectively), and also because they are composed of known mutations of interest or concern. The daily growth rate estimated for these variants by the logistic growth model was in the range of 2.44% to 7.18% (Fig 8C–8E).

The continued spread of some of these variants between February-April 2021 was confirmed by community-based PCR testing. To obtain independent frequency estimates, we made use of a PCR system sensitive to the presence of nsp6:Δ106–108 and S:Δ69–70 deletions (see Materials and Methods) to detect the B.1.1.7, B.1.1.524 and B.1.1.525 variants. Specifically, S:Δ69–70$^+$ nsp6:Δ106–108$^+$ samples correspond to B.1.1.7, while S:Δ69-70$^-$ nsp6:Δ106–108$^+$

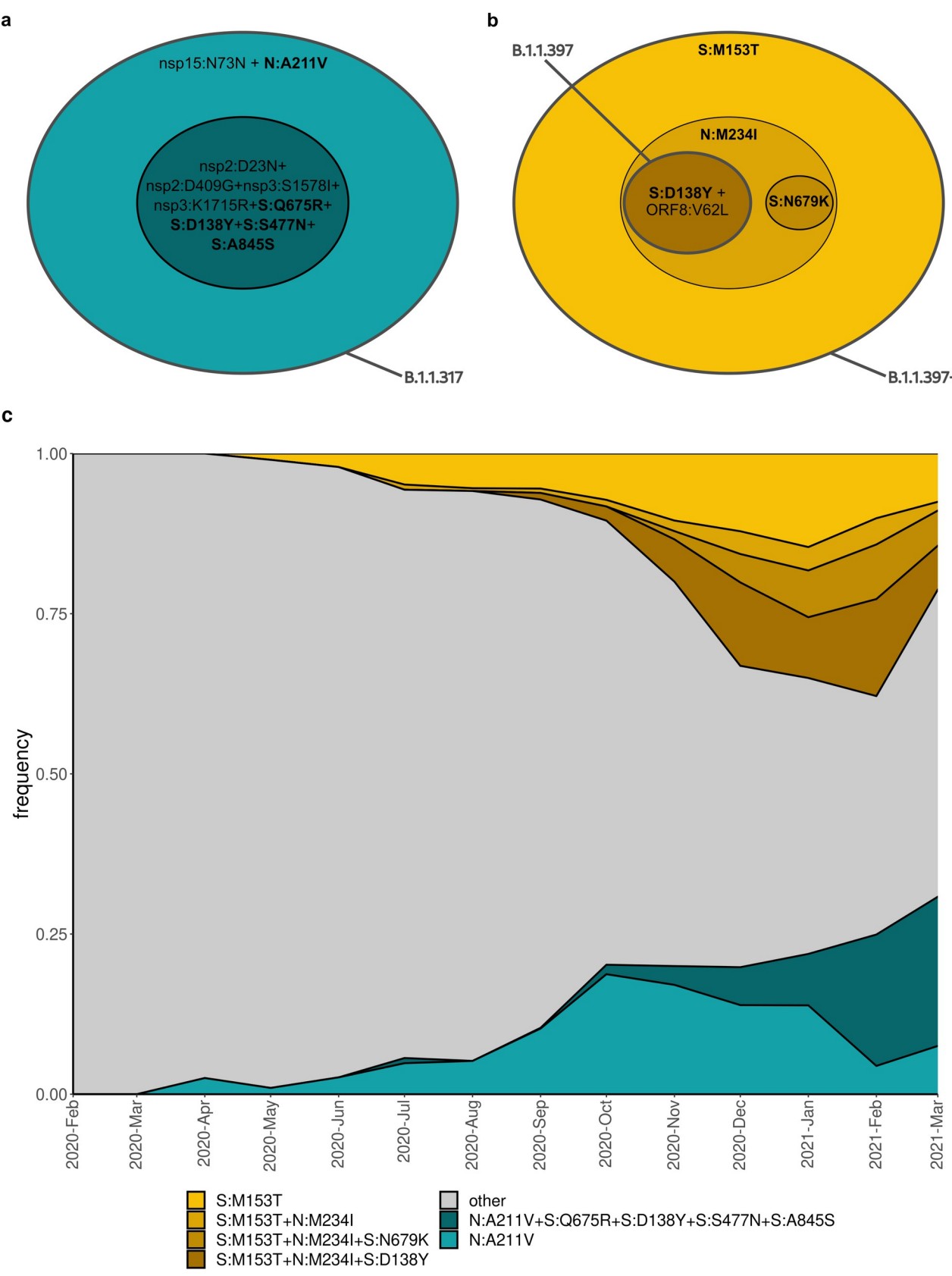

**Fig 5. Mutational composition and frequency dynamics of the B.1.1.317 and B.1.1.397+ lineages.** A, B: Schematic representation of the B.1.1.317 and B.1.1.397+ lineages. Pango lineage designations are approximate. C: Muller plots representing the frequency dynamics of the corresponding combinations of mutations in Russia.

samples correspond to either the B.1.1.524 or the B.1.1.525 variant (Fig 4). While the frequency estimates were highly uncertain (Table 1), they indicate that B.1.1.7, and one or both of variants B.1.1.524 and B.1.1.525, were wide-spread in April (Fig 9 and Table 1). A considerable fraction (59.6%) of PCR samples from February and March were included in our main analysis, as their sequences were in GISAID. However, the frequency increase was also observed in the 136 PCR samples for which no sequencing data was available (S2 Fig), providing independent validation of the NGS results. Similarly, it was observed when the PCR tests only for St. Petersburg were analyzed (S3 and S4 Figs), indicating that the prevalence of these variants increased at least in this city as opposed to being an artifact of changing sampling between regions.

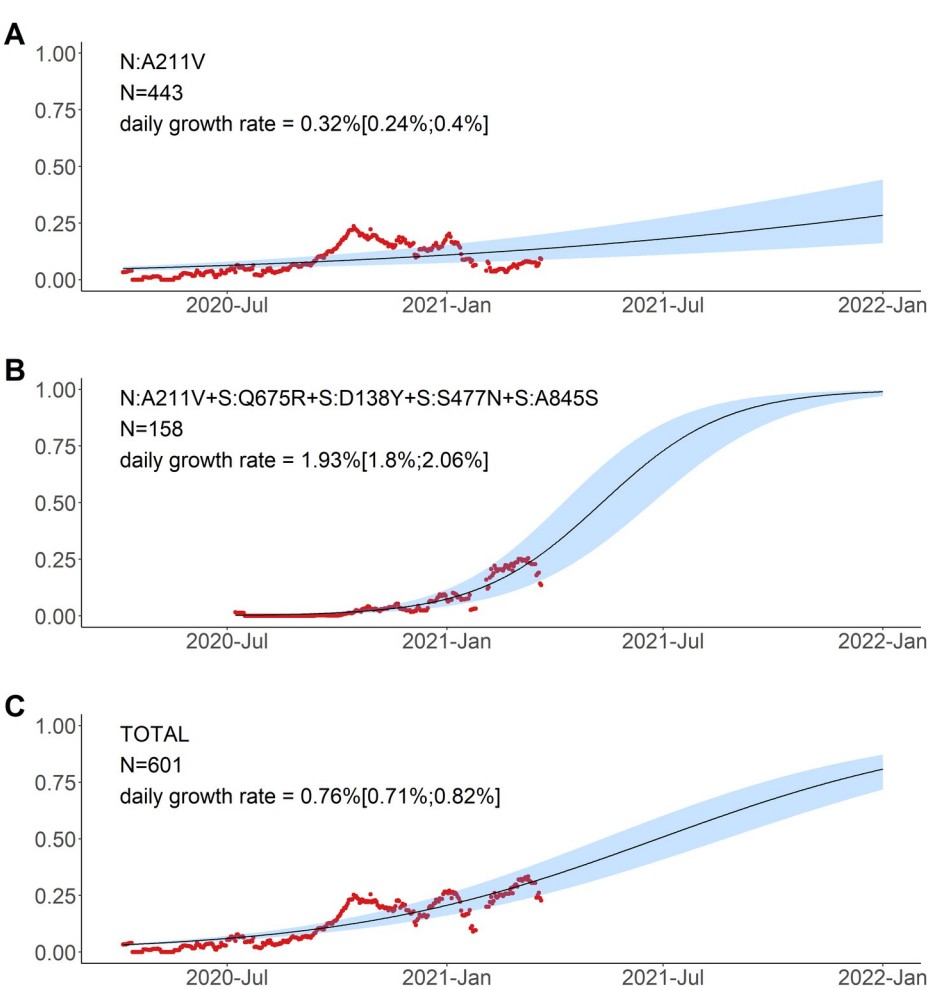

**Fig 6. Logistic growth model for nested variants defined by amino acid changes in the N:A211V context.** Red dots, sliding window 14-day average frequency; shaded area, 95% confidence interval. Variants are identified according to the presence of the mutations in S and N proteins; see Fig 5 for complete lists of mutations in the corresponding variants.

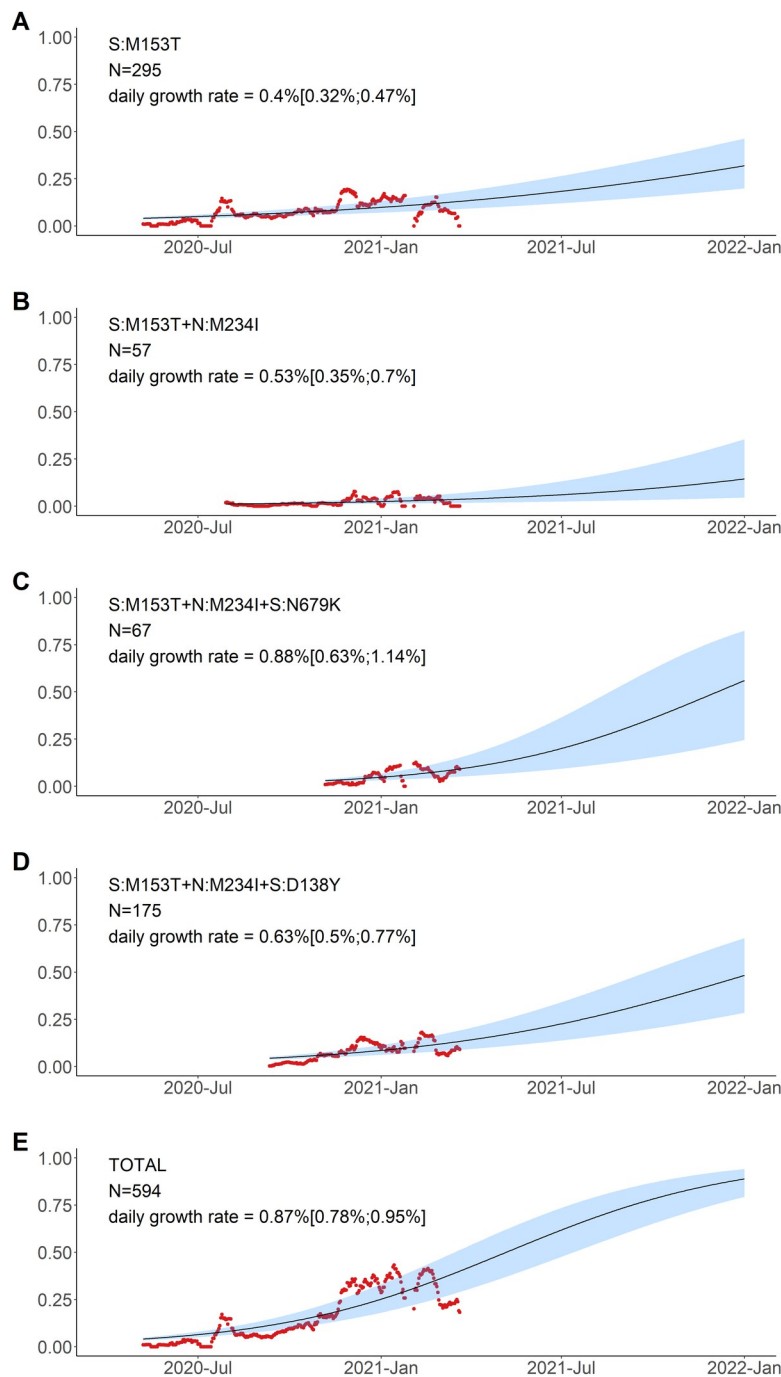

**Fig 7. Logistic growth model for variants defined by amino acid changes in the S:M153T context.** Notations are the same as in Fig 6.

## Mutational composition of the high-frequency variants

In this section, we discuss the mutations constituting the variants that were spreading in Russia before April 2021.

**B.1.1.317.** This lineage is defined by the presence of the N:A211V mutation. Changes at nucleocapsid position 211 experienced both persistent (according to the FEL model of HyPhy

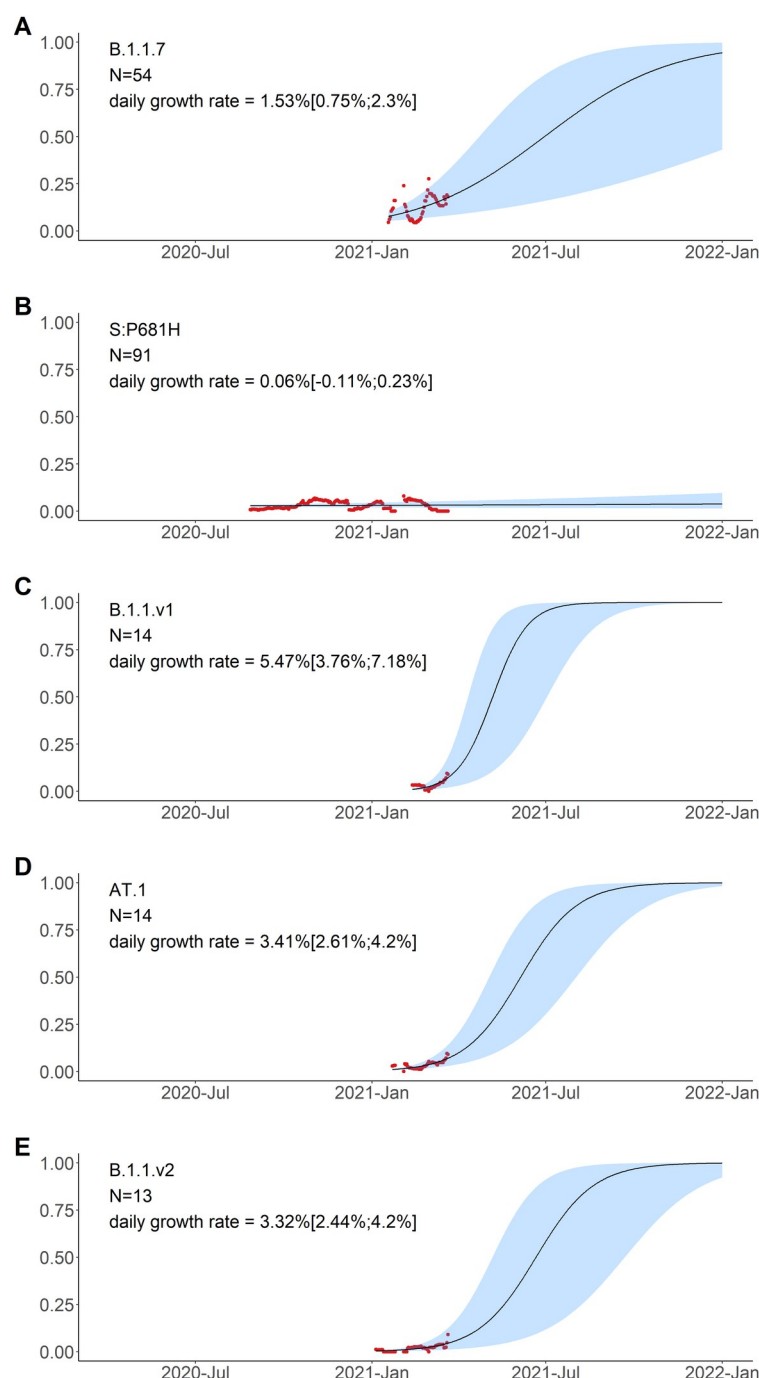

**Fig 8. Logistic growth model for the five remaining amino acid variants with high frequencies in Russia in February-March.** Notations are the same as in Fig 6. S:P681H has been observed both independently and as part of B.1.1.7; in panel B, the cases of B.1.1.7 are not shown, by excluding the S:P681H+nsp3:I1412T combination.

[20]) and episodic (according to the MEME model [21]) positive selection both in the global [29] and in the Russian dataset (p = 0.0396 for the MEME model and p = 0.0268 for the FEL model, the likelihood-ratio test), as well as a rapid increase in frequency of non-reference variants in the global dataset [29]. While the global frequency of 211V has remained low (<0.4%),

**Table 1. Frequencies of (B.1.1.524 or B.1.1.525) and B.1.1.7 estimated from PCR data.**

|  | 2021-Feb | 2021-Mar | 2021-Apr |
|---|---|---|---|
| nsp6:Δ106–108+ S:Δ69-70- (B.1.1.524 or B.1.1.525) | 10.5% (2.3%-31.4%) | 6.9% (4.1%-11.2%) | 15.2% (7.6%-28.2%) |
| nsp6:Δ106–108+ S:Δ69–70+ (B.1.1.7) | 10.5% (2.9%-31.4%) | 13.7% (9.7%-19.1%) | 21.7% (12.3%-35.6%) |

The point estimate and the 95% confidence intervals (Wilson score intervals) are shown.

in Russia, it has reached 26.9% in February-March 2021. According to immunoinformatic analysis, site N:211 is included in one of the four regions of the nucleocapsid protein with the highest affinity to multiple MHC-I alleles [30]. Nevertheless, the frequency of the variant carrying the N:A211V mutation alone has declined since October 2020 (Fig 5), suggesting that it is unlikely to confer transmission advantage against the background of other variants that were frequent in early 2021 (Fig 6). The B.1.1.317 lineage was detected in many countries in the end of 2020 and beginning of 2021, reaching tens of percent in some of them (S1 Table).

A subclade within B.1.1.317 that was spreading rapidly during the studied period carried the (Q675R+D138Y+S477N+A845S) combination of changes in spike. Two of these mutations are of interest. S:D138Y, first described as one of the lineage-defining mutations of the P.1 lineage [6], is a change in the N-terminal domain (NTD) of spike. Site 138 is adjacent to the NTD antigenic supersite, and together with other NTD mutations of P.1, S:D138Y was suggested to be the cause of disruption of binding with mAb159 [31] which is one of the most potent inhibitory antibodies [32]. S:S477N is positioned in the receptor-binding motif (RBM) of the S-protein near the antibody binding site (Fig 10) and was reported to promote resistance to multiple antibodies and plasma from convalescent patients [33,34]. Additionally, S477N is thought to increase ACE2 binding [35]. It is one of the lineage-defining mutations of the B.1.160 (20A.EU2) lineage prevalent in Europe in Autumn 2020 [36], and the only one among them to occur in the S-protein; it also defines one of the two subclades of the B.1.526 variant of interest that were spreading in the USA in the beginning of 2021 [37,38]. S:Q675R is located at the

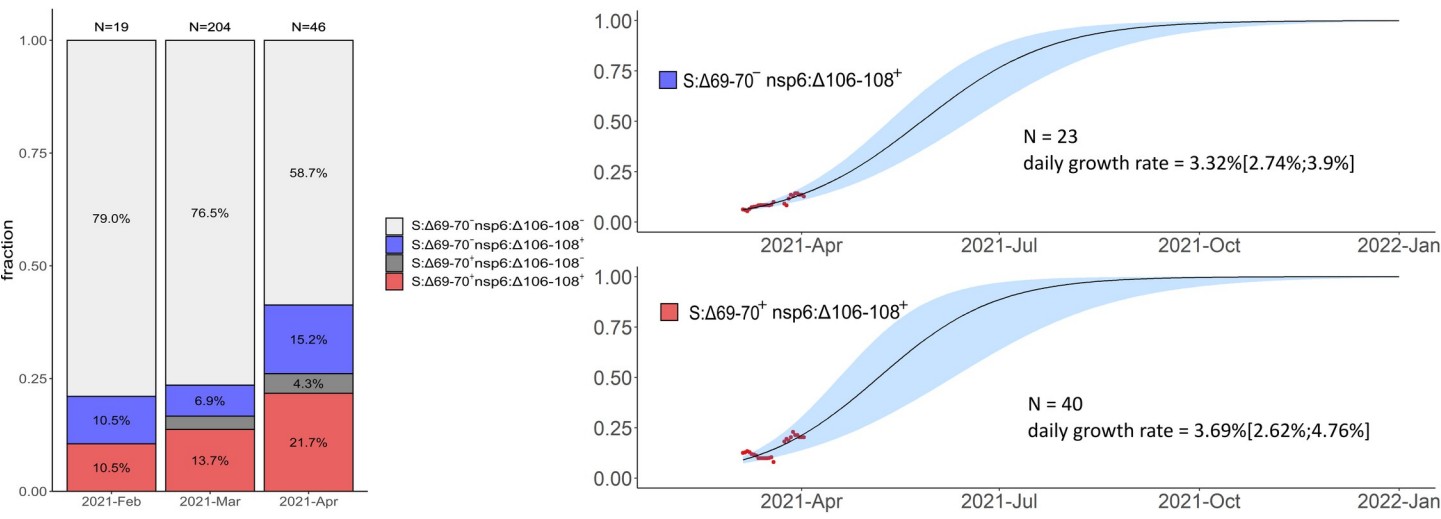

**Fig 9. Frequencies of S:Δ69–70, nsp6:Δ106–108, and their combination in Russia in Feb-Apr 2021 based on PCR data.** S:Δ69–70⁺ nsp6:Δ106–108⁺ samples correspond to B.1.1.7, and S:Δ69-70⁻ nsp6:Δ106–108⁺ samples correspond to B.1.1.524 or B.1.1.525. The rare instances of S:Δ69–70⁺ nsp6:Δ106-108⁻ probably correspond to false positive S:Δ69–70 detection. Notations for logistic curves are the same as in Fig 6.

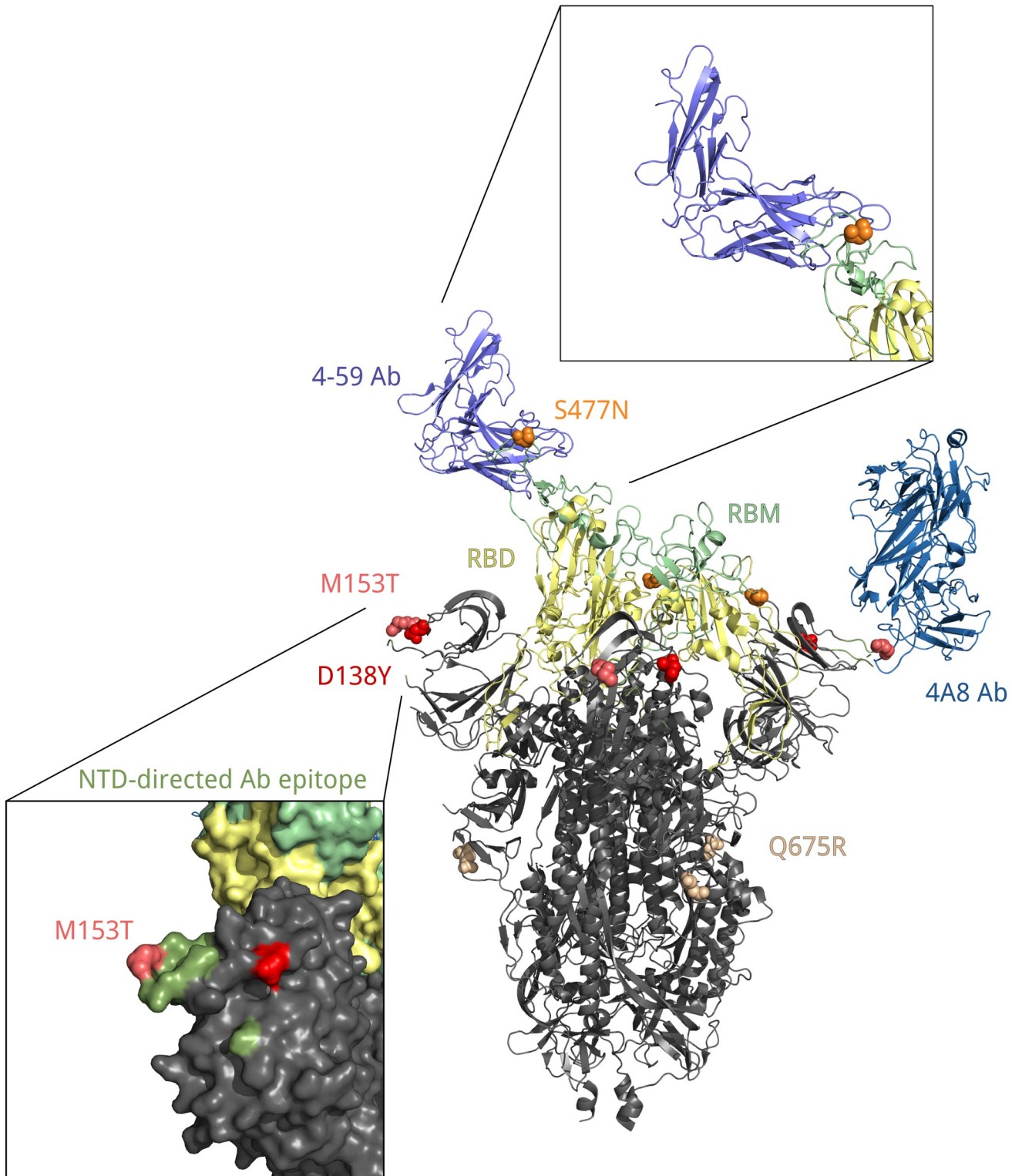

**Fig 10. Position of residues 138, 153, 477 and 675 in the spatial structure of the S protein bound with 4A8 and 4–59 antibodies (PDB IDs: 7c2l and 7czx).** Each of these residues is colored in its own color. Yellow, receptor binding domain; green, receptor binding motif; ocean blue, heavy chain of the 4A8 antibody; blue, heavy chain of the 4–59 antibody; olive, antibody binding epitope.

central part of S1 (Fig 10), and S:A845S, in S2; the significance of these two mutations is unknown.

**B.1.1.397+.** S:M153T is a characteristic mutation of B.1.1.397, which also comprises several other mutations. However, the frequency of S:M153T in Russia also increased in the absence of these other mutations (Fig 7). This increase has been ongoing since late spring 2020 (Fig 5), and has been noticed in Russia [39,40]. S:M153T, however, has remained rare outside Russia. S:153 is the first position of the 6-amino acid insertion specific to SARS-CoV-2 and some closely related bat betacoronaviruses that was absent in SARS-CoV [41]. While the effect of S:M153T on antigenic properties is unknown, S:153 is a part of the N3 loop of the NTD. This loop is a part of the NTD antigenic supersite (Fig 10; [42]), and nearby residues, including S:152, were recently shown to bind highly neutralizing 4A8 antibody from a convalescent patient [43]. Besides Russia, S:M153T was prevalent in several other countries including Kazakhstan and Mongolia (S2 Table) [19] which has a long border with Russia, suggesting common ancestry of this change in these countries.

The most frequent subclade within B.1.1.397+, and the one with some evidence for an independent increase in frequency (Fig 7), was defined by the presence of two additional mutations of interest: S:D138Y discussed above in the context of the B.1.1.317 lineage (but acquired in the B.1.1.397 lineage independently); and N:M234I. Position N:234 is a part of a disordered linker domain of the nucleocapsid protein [44]. Outside B.1.1.397, the N:M234I change has also occurred independently in several lineages that have attracted attention. It is among the lineage-defining mutations of the B.1.160 (20A.EU2) lineage as well as the B.1.526 lineage that increased in frequency in the USA at rates comparable to those of B.1.1.7 [37]. It was also one of the changes defining a lineage which also contained S:N501Y and S:P681H and seemed to spread rapidly in the USA [45]. Independent emergence of N:M234I in several variants of interest may suggest its impact on at least one of multiple functions of the N protein [46], although this can only be tested experimentally.

**Other notable variants.** The five other combinations of mutations observed at high frequencies in Russia in February-March 2021 were B.1.1.7, the best-known variant with increased transmissibility before the emergence of B.1.617.2 lineage; the variant carrying the S:P681H mutation alone; and three novel variants.

S:P681H is one of the nine spike changes that characterized B.1.1.7 lineage [4]; and S:P681R is one of the lineage-defining mutations of B.1.617.2 lineage; however, changes at this site are absent from two other lineages of concern, B.1.351 and P.1, indicating that it is not essential for increased transmissibility. The 681 position is adjacent to the furin cleavage site; this site is absent in non-human CoV, and is assumed to have contributed to pathogenicity in humans [47]. Changes at this position experienced both persistent and episodic positive selection [29]. P681H appeared to increase in frequency globally [48], although it was hard to disentangle this increase from that of the other changes constituting B.1.1.7 lineage that was rapidly spreading. We find that the frequency of this mutation in Russia in the absence of other B.1.1.7 mutations did not increase (Fig 8), indicating that it does not increase transmissibility by itself.

The three remaining high-frequency variants with evidence for rapid frequency increase in early 2021 carried combinations of the following high-frequency mutations: S:P9L, S:Δ140–142 (or S:Δ136–144), S:E484K, and nsp6:Δ106–108. The sets of mutations in these variants are in conflict (i.e., not nested within each other; Fig 4), indicating that at least some of these mutations emerged in them independently. These mutations are of interest or concern. Specifically, S:E484K (present in AT.1 and B.1.1.525) is involved in several famous variants including the B.1.351 [5], P.1 [6,49] and P.2 [6,49] lineages, and has been shown by several groups to cause escape from neutralizing antibodies [34,50,51]. nsp6:Δ106–108 (also referred to as ORF1a: Δ3675–3677, and present in B.1.1.524 and B.1.1.525) is a part of three previously circulating

variants of concern (B.1.1.7, B.1.351, P.1). S:Δ140–142 (present in B.1.1.524), S:Δ144 (present in B.1.1.525) and S:Δ136–144 (present in AT.1) are distinct deletions at a recurrent deletion region of the spike glycoprotein which confer resistance to neutralizing antibodies [52].

## Discussion

Russia has been relatively isolated in the course of COVID-19 pandemic: both the first cases of COVID-19 and the arrival of variants of concern, notably the B.1.1.7, have happened here later than in many European countries [27,53]. Together with the large size of the outbreak in Russia, such isolation could have created conditions for emergence of novel important domestic variants.

A steady increase in frequency of lineages B.1.1.317 and B.1.1.397+, as well as the presence of multiple mutations with potential effect on antigenic properties, notably S:D138Y, merit classification of these two lineages as variants of interest [54,55]. Nevertheless, there is no experimental data to suggest that these lineages have inherent properties that make them more transmissible. Furthermore, the rate of spread of B.1.1.317 and B.1.1.397+ has been lower than that of VOCs (e.g., ~7% for B.1.1.7 [16]). In particular, while B.1.1.317 has been observed in Russia since April 2020, the subclade of B.1.1.317 carrying the three spike mutations, since July 2020, and B.1.1.397+, since April 2020, all these lineages have remained at frequencies below 30% in Russia, and the logistic growth rates estimated by our model are all below 2% (Figs 6–8). Besides, these variants were missing spike changes L452R, E484K or N501Y which occur in other VOCs [55]. All these variants have been displaced with the advent of the Delta variant in summer 2021.

The three variants that emerged in 2021, AT.1, B.1.1.524 and B.1.1.525 (Fig 8C–8E), may be of more interest, because their estimated rate of frequency increase was higher and because they include mutations with known effects and occurring in other variants of interest or concern. While these variants have also become rare after the spread of the delta variant, the match between their mutational composition and that of other globally spreading variants of concern suggests that the mutations comprising them could have contributed to their rapid frequency increase. Importantly, this is just one of the possibilities. Again, there is no experimental data that demonstrates that these variants have different biological properties that make them more or less transmissible. By itself, a frequency increase is not a sufficient evidence for a fitness advantage of a variant, and some of the SARS-CoV-2 variants that have spread over the course of the pandemic confer no advantage in transmission [7]. Direct experimental studies are required to distinguish a transmission advantage conferred by mutations from epidemiological factors or random drift leading to variant spread.

## Supporting information

**S1 Fig. Frequency dynamics of SARS-CoV-2 amino acid changes in different regions of Russia.** Notations are the same as in Fig 2.
(PDF)

**S2 Fig. Logistic growth model for the S:Δ69–70+ nsp6:Δ106–108+ and S:Δ69-70- nsp6:Δ106–108+ samples in Feb-Apr 2021 based on the 136 samples for which no NGS data was available.** Notations are the same as in Fig 6.
(PNG)

**S3 Fig. Frequencies of S:Δ69–70, nsp6:Δ106–108, and their combination in Saint Petersburg in Feb-Apr 2021 based on the PCR data.**
(PDF)

**S4 Fig. Logistic growth model for nsp6:Δ106–108 with and without S:Δ69–70 in Saint Petersburg in Feb-Apr 2021 based on PCR data.** Notations are the same as in Fig 5.
(PNG)

**S1 Table. Maximum frequency of B.1.1.317 lineage in countries in samples per month.** For each country, only months with at least two samples were considered.
(XLS)

**S2 Table. Maximum frequency of B.1.1(B.1.1.\*) + S:M153T in countries in samples per month.** For each country, only months with at least two samples were considered.
(XLS)

**S1 File. The list of CORGI consortium members.**
(PDF)

**S2 File. Acknowledgments to GISAID.**
(ZIP)

## Acknowledgments

We are grateful to all GISAID submitting and originating labs (Supplementary File 2) for rapid open release of SARS-CoV-2 sequencing data and to all members of the CoRGI (Coronavirus Russian Genetic Initiative) Consortium that are listed in Supplementary File 1. We thank Sergei L Kosakovsky Pond for help with HyPhy analyses, and Evgeniya Alekseeva and members of the Bazykin lab for fruitful discussions.

## Author Contributions

**Conceptualization:** Galya V. Klink, Ksenia R. Safina, Georgii A. Bazykin.

**Data curation:** Andrey B. Komissarov, Dmitry Lioznov.

**Formal analysis:** Galya V. Klink, Ksenia R. Safina, Mikhail Moldovan, Georgii A. Bazykin.

**Funding acquisition:** Georgii A. Bazykin.

**Investigation:** Galya V. Klink, Ksenia R. Safina, Sofya K. Garushyants, Mikhail Moldovan, Elena Nabieva, Andrey B. Komissarov, Dmitry Lioznov.

**Supervision:** Georgii A. Bazykin.

**Visualization:** Galya V. Klink, Ksenia R. Safina, Sofya K. Garushyants.

**Writing – original draft:** Galya V. Klink, Georgii A. Bazykin.

**Writing – review & editing:** Galya V. Klink, Ksenia R. Safina, Elena Nabieva, Georgii A. Bazykin.

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
