## [Decision Letter · Decision Letter 0]

30 Mar 2022

PONE-D-22-00516Spread of endemic SARS-CoV-2 lineages in Russia before April 2021PLOS ONE

Dear Dr. Klink,

Thank you for submitting your manuscript to PLOS ONE. After careful consideration, we feel that it has merit but does not fully meet PLOS ONE’s publication criteria as it currently stands. Therefore, we invite you to submit a revised version of the manuscript that addresses the points raised during the review process.

Please refer to the additional editor comments. Looking forward to receiving a revised manuscript.

We look forward to receiving your revised manuscript.

Kind regards,

Luis M Schang, MV. Ph.D.

Academic Editor

PLOS ONE

Journal Requirements:

“This work was supported by the RFBR grant 20-54-80014 to G.A.B.”

“G.A.B.; grant 20-54-80014; Russian Foundation for Basic Research; https://www.rfbr.ru/rffi/eng.

3. One of the noted authors is a group or consortium CoRGI. In addition to naming the author group, please list the individual authors and affiliations within this group in the acknowledgments section of your manuscript. Please also indicate clearly a lead author for this group along with a contact email address

Additional Editor Comments (if provided):

Thank you for submitting your manuscript PLOS ONE. As you can see from the reviewers' comments, they both had a major concern regarding the discussion of growth advantages for alleles and variants in the absence of experimental evidence. Please, edit the manuscript thoroughly to limit the descriptions to the actual evidence. You may follow the suggestion of reviewer No.2 to add a paragraph to the discussion discussing this POTENTIAL mechanism for the spread of these particular alleles.

Reviewers' comments:

Reviewer's Responses to Questions

**Comments to the Author**

1. Is the manuscript technically sound, and do the data support the conclusions?

Reviewer #1: Yes

Reviewer #2: Partly

2. Has the statistical analysis been performed appropriately and rigorously? 

Reviewer #1: Yes

Reviewer #2: Yes

3. Have the authors made all data underlying the findings in their manuscript fully available?

Reviewer #1: Yes

Reviewer #2: Yes

4. Is the manuscript presented in an intelligible fashion and written in standard English?

Reviewer #1: Yes

Reviewer #2: Yes

5. Review Comments to the Author

Reviewer #1: The manuscript “Spread of endemic SARS-CoV-2 lineages in Russia before April 2021" by Klink et al. focuses on the spread of two SARS-CoV-2 lineages B.1.1.317 and B.1.1.397+ which were highly prevalent in Russia but rarely detected in sample outside of Russia. The authors speculate that specific nucleotide changes within the spike gene (including E484k), and not epidemiology, lead to the increase in the frequency of these lineages in Russia. Overall the data is well presented and the manuscript is interesting.

One issue is the authors’ suggestion that these variants have a transmission advantage or disadvantage, this is eluded to many time throughout the manuscript and is purely speculation. As there is no experimental data included which demonstrates that these variants have different biological properties that make them more or less transmissible, it is advisable that they remove or reword some of the speculation on transmissibility.

Minor points to clarify:

1. For the alignment to the original Wuhan sequence what was the length fraction and similarity fraction? How were nonspecific or repeats match handled?

2. The authors mention the global frequency of B.1.1.317 and B.1.1.397+ was low, and B.1.1.397+ isolates have been found in Kazakhstan. What other countries have B.1.1.317 and B.1.1.397+ isolates been identified?

3. Unclear what the authors mean by “suspicious” in the sentence (page 24, line 391-394): “The combinations of mutations seen in the three variants that emerged in 2021, AT.1, B.1.1.524 and B.1.1.525 (Fig. 8C-E), look more suspicious, because their estimated rate of frequency increase was higher and because they include mutations with known effects and occurring in other variants of interest or concern.”

4. The authors use the acronym VOC without defining it when it is used for the first time (page 8, line 83).

Reviewer #2: The submitted manuscript presents an analysis of the evolution of SARS-CoV-2 variants and lineage in Russia between February 25, 2020 and March 28, 2021, described according to the Pango lineage nomenclature. In brief, the B.1.1 lineage was preponderant, with the B.1..7, B1.1.317, and B.1.1.397 constituting the main sub lineages.Both geographical and temporal frequency distribution of several specific alleles are analyzed and logistic growth models are presented. The analyses are generally well and clearly presented, but some conclusions are overstated.

Major points.

In several instances, the authors discuss predicted fitness advantages of the alleles observed. Although this may well be correct, no actual experimental evidence is presented to substantiate these conclusions, which should therefore be removed. Wording of the entire manuscript should be carefully edited to avoid any claim of fitness advantages for alleles that have not been supported by experimental evidence. The discussion section may include a paragraph discussing this possibility, but founder and similar effects should not be dismissed without evidence.

6. PLOS authors have the option to publish the peer review history of their article (what does this mean?). If published, this will include your full peer review and any attached files.

Reviewer #1: No

Reviewer #2: No

---

## [Author Response · Author response to Decision Letter 0]

14 Jun 2022

Additional Editor Comments (if provided):

Thank you for submitting your manuscript PLOS ONE. As you can see from the reviewers' comments, they both had a major concern regarding the discussion of growth advantages for alleles and variants in the absence of experimental evidence. Please, edit the manuscript thoroughly to limit the descriptions to the actual evidence. You may follow the suggestion of reviewer No.2 to add a paragraph to the discussion discussing this POTENTIAL mechanism for the spread of these particular alleles.

Thank you, we now softened all statements about potential transmission advantage of considered lineages.

Reviewer #1: The manuscript “Spread of endemic SARS-CoV-2 lineages in Russia before April 2021" by Klink et al. focuses on the spread of two SARS-CoV-2 lineages B.1.1.317 and B.1.1.397+ which were highly prevalent in Russia but rarely detected in sample outside of Russia. The authors speculate that specific nucleotide changes within the spike gene (including E484k), and not epidemiology, lead to the increase in the frequency of these lineages in Russia. Overall the data is well presented and the manuscript is interesting.

We thank the Reviewer for the interest in our study.

One issue is the authors’ suggestion that these variants have a transmission advantage or disadvantage, this is eluded to many time throughout the manuscript and is purely speculation. As there is no experimental data included which demonstrates that these variants have different biological properties that make them more or less transmissible, it is advisable that they remove or reword some of the speculation on transmissibility.

We now carefully removed from the text all suggestions about the role of mutations in the transmissions of lineages, and left them only in the Discussion section. In Discussion, we make sure it is explicit that this is no more than a hypothesis.

Minor points to clarify:

1. For the alignment to the original Wuhan sequence what was the length fraction and similarity fraction? How were nonspecific or repeats match handled?

The average length of sequences was 29687.29 bp, and the average Hamming distance from the reference was 0.000799 (figures below). Therefore, the sequences had good coverage, and were reliable in that they did not have unexpectedly high divergence. Genomes were aligned with MAFFT using its default parameters for handling nonspecific matches. This is now stated in Methods.

2. The authors mention the global frequency of B.1.1.317 and B.1.1.397+ was low, and B.1.1.397+ isolates have been found in Kazakhstan. What other countries have B.1.1.317 and B.1.1.397+ isolates been identified?

Both B.1.1.317 and B.1.1.397+ have been found in many countries, but in most of them their frequencies remained low. We now added Tables S1 and S2 to show the prevalence of B.1.1.317 and and B.1.1.397+ in different countries in the month when this prevalence was the highest.

3. Unclear what the authors mean by “suspicious” in the sentence (page 24, line 391-394): “The combinations of mutations seen in the three variants that emerged in 2021, AT.1, B.1.1.524 and B.1.1.525 (Fig. 8C-E), look more suspicious, because their estimated rate of frequency increase was higher and because they include mutations with known effects and occurring in other variants of interest or concern.”

We mean that these lineages were to be considered as candidates for changed transmissibility, due to their mutational composition and a fast frequency increase. We now reformulated the phrase to “may be of more interest”. 

4. The authors use the acronym VOC without defining it when it is used for the first time (page 8, line 83).

Now we define VOC at first use, and give a link to the WHO website with this definition.

Reviewer #2: The submitted manuscript presents an analysis of the evolution of SARS-CoV-2 variants and lineage in Russia between February 25, 2020 and March 28, 2021, described according to the Pango lineage nomenclature. In brief, the B.1.1 lineage was preponderant, with the B.1..7, B1.1.317, and B.1.1.397 constituting the main sub lineages.Both geographical and temporal frequency distribution of several specific alleles are analyzed and logistic growth models are presented. The analyses are generally well and clearly presented, but some conclusions are overstated.

Major points.

In several instances, the authors discuss predicted fitness advantages of the alleles observed. Although this may well be correct, no actual experimental evidence is presented to substantiate these conclusions, which should therefore be removed. Wording of the entire manuscript should be carefully edited to avoid any claim of fitness advantages for alleles that have not been supported by experimental evidence. The discussion section may include a paragraph discussing this possibility, but founder and similar effects should not be dismissed without evidence.

We agree with the Reviewer and tone down all claims about any possible fitness advantage of considered lineages.

---

## [Decision Letter · Decision Letter 1]

16 Jun 2022

Spread of endemic SARS-CoV-2 lineages in Russia before April 2021

PONE-D-22-00516R1

Dear Dr. Klink,

We’re pleased to inform you that your manuscript has been judged scientifically suitable for publication and will be formally accepted for publication once it meets all outstanding technical requirements.

Kind regards,

Luis M Schang, MV. Ph.D.

Section Editor

PLOS ONE

Additional Editor Comments (optional):

Thank you very much for submitting a revised manuscript addressing all critiques to the original version

Reviewers' comments:

Reviewer's Responses to Questions

**Comments to the Author**

1. If the authors have adequately addressed your comments raised in a previous round of review and you feel that this manuscript is now acceptable for publication, you may indicate that here to bypass the “Comments to the Author” section, enter your conflict of interest statement in the “Confidential to Editor” section, and submit your "Accept" recommendation.

Reviewer #1: All comments have been addressed

2. Is the manuscript technically sound, and do the data support the conclusions?

Reviewer #1: Yes

3. Has the statistical analysis been performed appropriately and rigorously? 

Reviewer #1: Yes

4. Have the authors made all data underlying the findings in their manuscript fully available?

Reviewer #1: Yes

5. Is the manuscript presented in an intelligible fashion and written in standard English?

Reviewer #1: Yes

6. Review Comments to the Author

Reviewer #1: (No Response)

7. PLOS authors have the option to publish the peer review history of their article (what does this mean?). If published, this will include your full peer review and any attached files.

Reviewer #1: No

---

## [Editor Report · Acceptance letter]

24 Jun 2022

PONE-D-22-00516R1 

Spread of endemic SARS-CoV-2 lineages in Russia before April 2021 

Dear Dr. Klink:

I'm pleased to inform you that your manuscript has been deemed suitable for publication in PLOS ONE. Congratulations! Your manuscript is now with our production department. 

Kind regards, 

on behalf of

Dr. Luis M Schang 

Section Editor

PLOS ONE